# Do Mental Health and Vitality Mediate the Relationship between Perceived Control over Time and Fear of COVID-19? A Survey in an Italian Sample

**DOI:** 10.3390/jcm10163516

**Published:** 2021-08-10

**Authors:** Silvana Miceli, Barbara Caci, Michele Roccella, Luigi Vetri, Giuseppe Quatrosi, Maurizio Cardaci

**Affiliations:** 1Department of Psychology, Educational Science and Human Movement, University of Palermo, 90121 Palermo, Italy; silvana.miceli56@unipa.it (S.M.); michele.roccella@unipa.it (M.R.); maurizio.cardaci@unipa.it (M.C.); 2Department of Health Promotion, Mother and Child Care, Internal Medicine and Medical Specialties (ProMISE), University of Palermo, 90121 Palermo, Italy; luigi.vetri@gmail.com (L.V.); giuseppe.quatrosi01@community.unipa.it (G.Q.)

**Keywords:** perceived control over time, COVID-19, COVID-19 fear, mental health, vitality, health, quality of life, anxiety, emotion

## Abstract

Several studies evidenced increased elevated symptomatology levels in anxiety, general stress, depression, and post-traumatic stress related to COVID-19. Real difficulties in the effective control of time that could be responsible for mental health issues and loss of vitality were also reported. Prior literature highlighted how perceived control over time significantly modulates anxiety disorders and promotes psychological well-being. To verify the hypothesis that perceived control over time predicts fear of COVID-19 and mental health and vitality mediate this relationship, we performed an online survey on a sample of 301 subjects (female = 68%; M_age_ = 22.12, SD = 6.29; age range = 18–57 years), testing a parallel mediation model using PROCESS macro (model 4). All participants responded to self-report measures of perceived control over time, COVID-19 fear, mental health, and vitality subscales of the Short-Form-36 Health Survey. Results corroborate the hypotheses of direct relationships between all the study variables and partially validate the mediation’s indirect effect. Indeed, mental health (a1b1 = −0.06; CI: LL = −0.11; UL = −0.01; *p* < 0.001) rather than vitality (a2b2 = −0.06; CI: LL = −0.09; UL = 0.03; n.s.) emerges as a significant mediator between perceived control over time and fear of COVID-19. Practical implications of the study about treatment programs based on perceived control over time and emotional coping to prevent fear and anxiety toward the COVID-19 pandemic are discussed.

## 1. Introduction

Since its first identification by the Wuhan Municipal Health Commission in China, the 2019 SARS-CoV-2 pandemic has now reached more than one year after its spread and assumed more dramatic proportions in the entire world ever. Most countries experienced second waves and perhaps prepared for even a third related to the SARS-CoV-2 virus mutations. As of March 2020, the excess mortality in Italy was 20.4%, lower than that of Spain (23.6%), Belgium (20.8%), and Poland (23.2%) but higher than that of France (13.2%), Germany (7%), Holland (14.7%), and Portugal (13.9%) (ISTAT: 2021). The world’s social-economic system is in crisis, and the health system is overloaded. The latter must continuously deal with the coronavirus health emergency and is called upon to manage the increase in mental disorders deriving from the pandemic’s living conditions. Several studies have analyzed the negative consequences of the coronavirus pandemic on people’s mental health especially in patients with chronic or autoimmune diseases [1,2,3], evidencing elevated symptomatology levels in anxiety, general stress, depression, and post-traumatic stress related to COVID-19 [4]. Unpleasant emotions, anger, or vulnerability [5]; loss of vitality [6]; a lack of energy, an inability to start and carry out daily activities, or difficulty concentrating at work [7]; fatigue [8]; and social media addiction [9,10] are other negative consequences of the COVID-19 pandemic reported by the current literature. The coronavirus pandemic urgently forces people to live in the present moment, experiencing a sort of temporal disintegration in which time is stopped or slowed, the order of time and days confused. Even the future seems shortened, causing a real difficulty in the effective control of time in many cases. Therefore, this altered time control could be responsible for higher psychological distress and mental health issues [11]. Prior studies highlighted how perceived control over time significantly affects anxiety disorders [12]. It has also evidenced that it promotes psychological well-being [13] and coping strategies [14]. High levels of perceived control over time are associated with low levels of stress, high academic performance levels, problem-solving ability, and health [14].

Starting from these premises, we first aim to analyze the effect of perceived control over time on COVID-19 fear, a new form of situational anxiety [15]. We assume that higher levels of perceived control over time, which allow people to manage daily activities, are related to lower levels of COVID-19 fear. Therefore, Hypothesis 1 arises as follows:

**Hypothesis 1 (H1)**.*Perceived control over time is negatively associated with COVID-19 fear*.

Another essential aspect of perceived control over time is the socio-cognitive-behavioral attribute of perceived control. Perceived control has been defined in the psychological literature as personal control, locus of control, self-efficacy, and sense of control, even if each term has distinct features [16,17,18]. In the framework of social learning theory [16], perceived control over time is a predictor of future health behavior and status. In general, socio-cognitive psychologists suggested that high levels of perceived control are related to proactive behaviors and the ability to feel healthy. In contrast, low levels are associated with depression, stress, and anxiety-related disorders [19]. Longitudinal and reciprocal relationships between perceived control over life circumstances and mental health have been suggested too [8]. In this sense, perceived control modulates affective responses to environmental stressors, enhances positive emotions, decreases negative ones, and sustains psychological vitality [20]. People with high vitality are more active and productive and have good coping strategies and greater well-being [21,22,23,24]. Vitality was also related to psychological distress [25], subjective happiness [23], and physical function and health-related quality of life [25,26].

Hence, the second aim of the current study is to explore the reciprocal relationships among perceived control over time, mental health, and vitality. In the framework of the coronavirus pandemic, we hypothesize that people with a high sense of control over their time, who can plan their working or studying tasks without procrastination, will be more able to manage their negative emotions and increase their sense of vitality compared to those with low control beliefs over time. Consequently, we state Hypotheses 2 and 3 as follows:

**Hypothesis 2 (H2)**.*Perceived control over time is negatively associated with adverse mental health statuses*.

**Hypothesis 3 (H3)**.*Perceived control over time is positively associated with vitality*.

Considering the linear relationships evidenced by literature for COVID-19 anxiety and mental health and vitality, as described above [4], we expect to find that people with both adverse mental health status and low vitality levels will present higher levels of COVID-19 fear as well.

We hypothesize then that

**Hypothesis 4 (H4)**.*Adverse mental health statuses are positively associated with COVID-19 fear*.

**Hypothesis 5 (H5)**.*Vitality is negatively associated with COVID-19 fear*.

Finally, in the current study, we tested a model assuming mental health and vitality as mediators in the relationship between perceived time control and COVID-19 fear. Prior psychological studies reported mental health effects on increasing well-being and performance [27], decreasing anxiety, heightening self-confidence, and improving individuals’ self-control performance [28]. Scholars also showed that vitality contrasts with physical and mental fatigue, one of the pandemic’s main psychological consequences [8]. Although the time available to us to carry out our activities has dramatically increased during the imposed restrictions, individuals often experienced a lack of energy, an inability to start and manage daily activities, and difficulty concentrating at work [7]. Such a state of fatigue can adversely affect a person’s physical and mental well-being, but also, if prolonged, can, in the long term, predispose the individual to the onset of psychiatric diseases, especially depression. Morgul et al. [29], out of 4700 Turkish-nationality subjects, highlighted how fatigued individuals had more pessimistic attitudes toward the possibility that COVID-19 will finally be controlled, less satisfaction with the authorities’ preventive measures, and less confidence that their country can overcome the COVID-19 pandemic. They also showed low compliance with safety protocols such as wearing masks, washing hands, and maintaining physical distance. Consequently, adverse mental health status and loss of vitality may decrease perceived control over time and increase anxiety toward COVID-19.

We then formulate Hypothesis 6:

**Hypothesis 6 (H6)**.*Mental health and vitality mediate the relationship between perceived control over time and COVID-19 fear*.

In sum, the present study’s results would contribute to studying the dramatic effects of the coronavirus pandemic state on people’s anxiety levels, evidencing mental health and vitality as protective factors against anxiety toward SARS-Cov-2 infection.

## 2. Materials and Methods

### 2.1. Participants

A sample of 301 subjects (female = 68%; *M*_age_ = 22.12, SD = 6.29; age range = 18–57 years), coming almost all from southern Italy (85.7%), and having a diploma (79.1%) or a degree (16.3%) took part in the survey. According to the Declaration of Helsinki, all participants gave written consent about the anonymity of data handling, and the Bioethics Committee of the University of Palermo approved the study (n. 2/2020).

### 2.2. Procedure

Participants were recruited by responding voluntarily to the survey administered online via Google Forms on the researchers’ distance learning university courses during Italy’s lockdown phase. A snowballing procedure, asking subjects to recruit future participants from among their acquaintances, combined with respondent-driven sampling was used for acquiring a representative sample of the general population [30]. Google Forms presented the study information sheet in the first section. Data were automatically collected when participants filled the Google form online, reporting the electronic version of the assessment instrument consisting of demographic questions (i.e., gender, age, and education) and apposite measures of the studied variables. On average, the respondents completed the survey in 30 min.

### 2.3. Instruments

#### 2.3.1. Perceived Control over Time

The perceived control over time subscale from the Time Management Behavior Scale of Macan et al. [31] was used in the current study to measure the individuals’ perception of control over their time. It consists of five items scored on a five-point Likert scale with anchors from 1 = strongly disagree to 5 = strongly agree (example of item:” I feel in control of my time”; “I must spend much time on unimportant tasks”). The scale provides a total score by averaging the participants’ scores for each scale item (Cronbach α = 0.70).

#### 2.3.2. Fear of COVID-19

Fear of COVID-19 is a recent seven-item scale developed by Ahorsu et al. [15] to measure the fear of COVID-19 in the adult population (example of item: “I am most afraid of coronavirus-19”). Each item is scored on a five-point Likert scale having anchors from 1 = strongly disagree to 5 = strongly agree. We computed the total score by averaging the participants’ scores for each scale item (Cronbach α = 0.86).

#### 2.3.3. The Short-Form-36 Health Survey (SF-36)

We used both the five-item mental health and the four-item vitality subscale of the Italian version of the Short-Form-36 Health Survey [32], a general measure of population distress. One example of a mental health subscale item is “Have you felt so down in the dumps that nothing could cheer you up?”. One item example of the vitality subscale is “Did you feel tired?”. Each item is scored on a five-point Likert scale having anchors from 1 = strongly disagree to 5 = strongly agree. We computed the total score by averaging the participants’ scores for the mental health scale (Cronbach α = 0.80) and for the vitality scale (Cronbach α = 0.75).

### 2.4. The Parallel Mediation Model

A parallel mediation model was used in the study to verify our assumptions. It is a basic mediation model (4) from Hayes PROCESS templates [33]. Our theoretical model (Figure 1) assumes that perceived control over time (X) would indirectly affect fear of COVID-19 (Y) through two mediators: mental health (m1) and vitality (m2). All the individual direct and indirect paths as the total indirect effect were calculated as described in Figure 1.

### 2.5. Data Analysis

All the analyses were conducted through SPSS version 24 (IBM, Chicago, IL, USA). The first step was to calculate descriptive statistics and zero-order correlations. In the preliminary analysis, data were checked for accuracy. It was found that there were no missing values in the data. A correlation was used to see the relationship among all the included variables. The Hayes’s PROCESS macro was used to test our mediational model since it is considered a more powerful and effective method than its alternatives [33]. Before testing the model, all variables were standardized. The parameters were estimated using the bootstrap method with 5000 samples and a 95% confidence interval (CI) using the percentile method bias corrected [31]. The parameters are significant if the CI does not include zero.

## 3. Results

### 3.1. Descriptive Statistics

Table 1 shows descriptive statistics, mean values, and standard deviations and a correlation matrix for the study variables.

On average, participants reported medium–high scores in the perceived control over time scale (*M* = 3.32; SD = 0.70), low scores in the fear of COVID-19 scale (*M* = 1.76; SD = 0.69), and medium–low scores in SF-36 mental health (*M* = 2.74; SD = 0.84) and vitality subscales (*M* = 3.0; SD = 0.83). These results indicate that our sample can manage and control their time during the COVID-19 pandemic but are slightly scared by the SARS-CoV-2 virus. Participants also revealed that the COVID-19 pandemic has a medium impact on their mental health and vitality. A series of paired samples t-test indicated that there was a statistically significant difference, t(298) = −2.27, *p* < 0.01 on perceived control over time scores despite females (*M* = 3.39, SD = 0.70) obtaining higher scores than males (*M* = 3.31, SD = 0.60). Data also shows a statistically significant difference between COVID-19 scores, t(298) = −4.77, *p* < 0.01, with females (*M* = 1.88, SD = 0.70) reporting higher scores then males (*M* = 1.49, SD = 0.49) and depicting themselves as significantly more scared about SARS-CoV-2 infection. No significant differences between mental health and vitality scores in males and females were found.

### 3.2. Correlation and Regression Analyses

The correlation matrix exhibits a significant association between perceived control over time and fear of COVID-19 and the mediators under study—i.e., mental health and vitality. The correlation coefficient for perceived control over time and COVID-19 fear was −0.12 (*p* < 0.05), supporting our assumption H1 about the linear relationship between X and Y (X→Y) that perceived control over time is related to COVID-19 fear. Similarly, the correlation between perceived control over time and mental health (path a1) was significant with the coefficient r = 0.33 (*p* < 0.01) and that between perceived control over time and vitality (path a2) was r = 0.38 (*p* < 0.01), supporting hypotheses H2 and H3 of an association between our predictor (i.e., the perceived control over time) and mediators (i.e., mental health and vitality).

On the other hand, there was a positive relationship between mental health and COVID-19 fear (path b1) with r = 0.25 (*p* < 0.01), whereas vitality and COVID-19 fear (path b2) were negatively associated with r = −0.22 (*p* < 0.01), supporting H4 and H5 of significant linear relations between mediators and the dependent variable. Baron and Kenny [34] recommended that mediators be significantly associated with independent and dependent variables.

It must be noted that we found a significant negative correlation among gender (dummy coded with 1 = male; 0 = female) and perceived control over time (r = −0.13, *p* < 0.05) or fear of COVID-19 (r = −0.26, *p* < 0.01), showing females reported higher scores on the scale measuring perceived control over time and had a higher fear of COVID-19 than males. Furthermore, a significant negative relationship was found between gender and adverse mental health (r = −0.15; *p* < 0.01), whereas a positive, significant one has emerged between gender and vitality (r = 0.15; *p* < 0.01). Therefore, females present higher adverse mental health statuses and lower vitality scores than males.

All data were then examined through regression to judge whether to include them in the path model or in order to establish the study variables’ significance. After analyses, it was found that both mediators could be included in the path model. The independent variable explained 25% of the variance on its own, F = 7.09, R-square = 0.06, *p* < 0.001. Hence, all our assumptions (H1–H5) were supported.

### 3.3. The Mediation Role of Mental Health and Vitality on the Relationship between Perceived Control over Time and Fear of COVID-19

To test the model in which mental health and vitality mediate the relationship between perceived control over time and fear of COVID-19, a parallel mediation analysis using PROCESS model 4 was conducted on data from the whole sample.

Table 2 describes all the direct and indirect effects.

### 3.4. Direct Effects

Separate regression models were estimated for the entire path. First, as reported in Figure 2, mental health, the first mediator, was regressed on perceived control over time (path a1). Results show a significant negative association between perceived control over time and mental health, evidencing people with low perceived control over time having adverse mental health statuses. Secondly, the other mediator, vitality, was regressed on perceived control over time (path a2), and we found a significant positive path between perceived control over time and vitality. Thus, people with high perceived control over time also have an increased sense of vitality.

Similarly, in the model, fear of COVID-19 was regressed on mental health (path b1). It was found that there was a significant positive effect, whereas, when fear of COVID-19 was regressed on vitality, we found a significant negative path (b2). The direct effect was explored by regressing the fear of COVID-19 on perceived control over time, which was not significant. This result allows us to verify for a total mediation.

### 3.5. Indirect Effects

The regression model predicts fear of COVID-19 from mental health, vitality, and perceived control over time. Through all the mediators, we can see a strong negative effect both for mental health (a1b1 = −0.06; CI: LL = −0.11; UL = −0.01) and for vitality (a2b2 = −0.06; CI: LL = −0.09; UL = 0.03). This result shows that only adverse mental health is significantly associated with perceived control over time and fear of COVID-19, considering that bootstrap CI is above zero while controlling for demographic variables.

### 3.6. Total Effects

The direct effect of perceived control over time on fear of COVID-19 (c’) was −0.03 (CI: LL = −0.15; UL = 0.08: *p* = n.s.) (see Figure 2). In contrast, the total indirect effect via both mediators (a1b1 + a2b2) was −0.12 (CI: LL = −0.23, UL = −0.01, *p* < 0.001). Consequently, the total effect (a1b1 + a2b2 + c’) of X on Y was −0.15. Therefore, the total effect (c = −0.12; CI: LL = −0.23, UL = −0.01, *p* < 0.05) of perceived control over time on fear of COVID-19 was due to a negative indirect path, as the coefficient for direct effect (c’ = −0.03) was higher than that of the total indirect effect (−0.15). Hence, results show that mental health is a significant mediator between perceived control over time and fear of COVID-19 whereas vitality is not.

## 4. Discussion

The present study has multiple aims: the first goal is to determine the predictive role of perceived control over time on COVID-19 fear; the second is to analyze the reciprocal linear associations among perceived control over time, mental health, and vitality as well those among mental health, vitality, and COVID-19 fear; the final aim is to understand the influence of perceived control over time on fear of COVID-19 by considering how mental health and vitality mediate this relationship.

The whole sample shows a significant negative association between perceived control over time and fear of COVID-19, corroborating H1 (path c). It should be noted that the subscale of perceived control over the time of the Time Management Behavior Scale of Macan et al. [18], which we used in the present study, explored how perceived control over time affects task completion within deadlines or procrastination. Thus, in line with prior literature reporting that perceived control may act as a protective factor buffering the psychological impact of the pandemic on general health and life satisfaction [35], we found that during the COVID-19 lockdown, when people feel confident in managing their time effectively, they also have less anxiety [36]. We could hypothesize that even if people are forced to remain at home by the lockdown restriction, they could be less pressured by the so-called deadline rush in completing their work, studying, or completing familiar tasks [37]. Thus, their belief in their ability to control their time prevents them from experiencing COVID-19 fear. The present study also evidences gender differences in perceived control over time and COVID-19 fear, showing females having higher scores than males. This outcome is not contradictory. Indeed, even if our data come from a sample composed chiefly of young undergraduate females (female = 68%), they reflect the prior findings in the literature on perceived control, usually depicting young and highly educated women as more able to manage their daily activities than men [34]. In addition, it is in line with psychological literature reporting that women having higher levels of chronic and daily stressors [20] and higher emotional vulnerability, particularly when responding to negative emotions in comparison with men [38]. It also must be noted that females did not differ from males in their scores for mediators.

We secondarily found that perceived control over time is significantly related to mental health and vitality, so data fully support H2 (path a1) and H3 (path a2). Research has demonstrated that people’s ability/inability to control time is associated with well-being [13] and mental health complaints [39,40]. Our findings show a significant negative relationship between people’s time management beliefs and mental health, indicating that individuals with higher perceived control over time had lower negative mental health statuses. Indeed, the recent literature on the COVID-19 pandemic assigned an essential role to perceived control in moderating the relationship between the perceived severity of COVID-19 and mental health problems. In particular, the more people perceive themselves to be in control of their lives (and time), the more satisfied and healthier they were [41].

Third, our results find strong evidence of the significant positive relationship between fear of COVID-19 and adverse mental health statuses and, conversely, of the significant negative relationship between fear of COVID-19 and vitality, fully corroborating H4 (path b1) and H5 (path b2). From this perspective, our results are consistent with recent outcomes showing that the COVID-19 fear represents a sort of chronic mental anxiety perceived by individuals under the current uncertain and ongoing pandemic [42,43,44,45]. This psychological distress leads people to be continuously worried about their health and future infection by SARS-CoV-2: depression, anxiety, boredom, loneliness, insomnia, or anger symptoms [46] are reported; in turn, adverse mental health statuses and loss of vitality enhance the fear toward COVID-19 infection.

Finally, we found partial support for H6 (path c’) since the present study shows that only mental health but not vitality significantly mediates the relationship between perceived control over time and fear of COVID-19. We tested a parallel mediation model considering the effect of both mental health and vitality. We need to focus on the items’ content measured by the SF-36 mental health and vitality subscales to understand this outcome. Indeed, on the one hand, the mental health subscale asks people to indicate what kind of emotions they feel, e.g., nervous, happy, peaceful, sad, or down in dumps; on the other hand, the vitality subscale requires people to indicate their level of energy, tiredness, and fatigue. Although the recent literature reported how fear of COVID-19 affects vitality [29] in line with the theory of perceived control, our results corroborate that ability to handle environmental stress is closely associated with emotional state. Controlling time alters individuals’ perceptions regarding threats [47]. In turn, the crucial mediational role of mental health we found in the present study is supported by prior psychological literature about emotion regulation, demonstrating how emotions are critical factors for guiding individuals to avoid or cope with distressing situations, bringing about a positive outcome [48]. Therefore, we could argue that in managing such a negative situation as the COVID-19 pandemic, the loss of vitality represents more a physiological state related to inactivity and social isolation. Mental health statuses associated with emotions could, in contrast, vary during the day. Thus, even while experiencing exhaustion, positive or negative emotions are essential in modulating the relationship between perceived control over time and the fear of being infected. When people have favorable feelings, they also have a better perception of their time, manage their daily routines or working duties, and are less anxious toward the COVID-19 pandemic. Such a result has practical implications for developing treatment programs to enhance persons’ capacity and competence to handle their daily activities, schedule their time, and use effective techniques based on emotional coping to prevent fear and anxiety toward the COVID-19 pandemic.

However, the present study has some limitations to report. First, the sample is based mainly on an Italian university population; thus, we cannot generalize to the entire general population. Further studies should be performed, for instance, on samples of workers or professionals to corroborate our results. A second limitation is the study’s cross-sectional design, based on a sample mainly comprising women. Future studies should be based on a longitudinal design to compare the mental health and vitality statuses in the different lockdown phases and obtain a more balanced sample with regard to gender.

## 5. Conclusions

As suggested by the World Health Organization, mental health represents a public health priority [49], and it has become more significant nowadays due to the current COVID-19 pandemic. In sum, our study evidences the crucial mediating role of mental health between perceived control over time and COVID-19 anxiety. We measured mental health using the SF-36 mental health subscale assessing positive and adverse emotions. Hence, mental health could influence the perceived control over time, emotionally regulating people’s behaviors in different life domains, such as for instance academic achievements [50], job performance [51], or social activities [52], improving well-being and quality of life. Thus, mental health also becomes a protective factor from psychopathological threats related to COVID-19 fear. Furthermore, our results have practical implications since they could help psychologists, psychiatrists, and educators define treatment programs and guide the states’ governance measures combating psychological distress related to the COVID-19 pandemic.

## Figures and Tables

**Figure 1 jcm-10-03516-f001:**
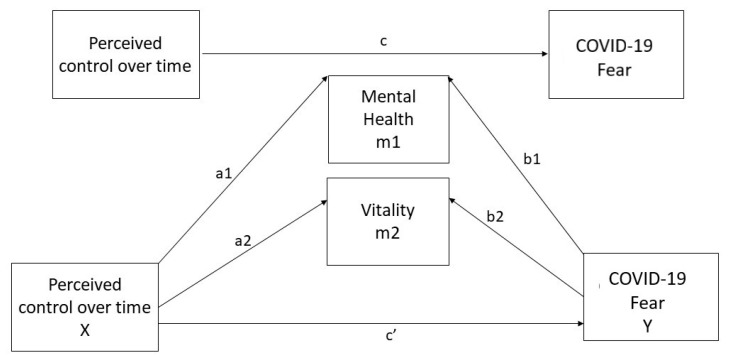
Hypothesized model. Note—perceived control over time: predictor variable; mental health: mediator 1; vitality: mediator 2; COVID-19 fear: outcome variable; a1b1: a specific indirect effect of perceived control over time on COVID-19 fear through mental health; a2b2: a specific indirect impact of perceived control over time on COVID-19 fear through vitality; c’: direct effect of perceived control over time on COVID-19 fear; total indirect effect: a1b1 + a2b2.

**Figure 2 jcm-10-03516-f002:**
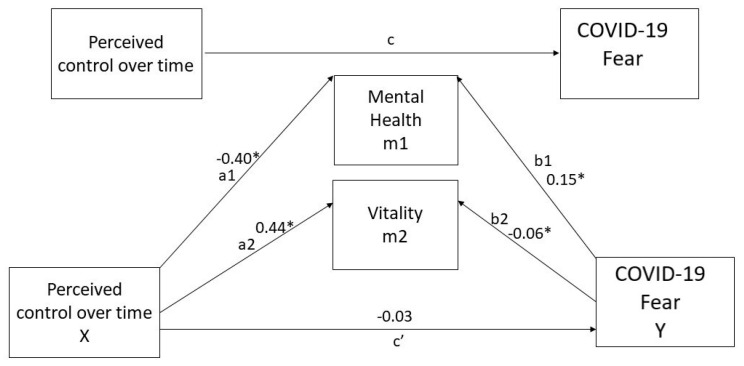
The mediating effect of mental health dimensions in the relationship between perceived control over time and fear of COVID-19. Note: * *p* < 0.001. All the presented effects are unstandardized.

**Table 1 jcm-10-03516-t001:** Descriptive statistics and zero-order correlation matrix.

	Mean (SD)	1	2	3	4	5	6
1. COVID-19 fear	1.76 (0.69)	1					
2. Perceived control over time	3.32 (0.70)	−0.12 *	1				
3. Vitality	3.00 (0.83)	−0.22 **	0.38 **	1			
4. Mental health	2.74 (0.84)	0.25 **	−0.33 **	−0.72 **	1		
5. Gender (dummy coded: male = 1; female = 0)		−0.26 **	−0.13 *	0.15 **	−0.15 **	1	
6. Age		−0.03	0.03	−0.00	−0.01	0.03	1

* Correlation is significant at *p* < 0.05 (two tiles). ** Correlation is significant at *p* < 0.01 (two tiles).

**Table 2 jcm-10-03516-t002:** Path coefficients for parallel mediation model.

Path	Effect	Boot LL-CI	Boot UL-CI	SE	T	*p*-Value
Total effect	−0.12	−0.23	−0.01	0.05	−2.20	0.028
Direct effect (c’)	−0.03	−0.15	0.08	0.05	−0.59	0.555
IV–m1 (a1)	−0.40	−0.53	−0.27	0.06	−6,.22	0.000
IV–m2 (a2)	0.44	0.32	0.57	0.06	7.17	0.000
M1–DV (b1)	0.15	0.01	0.28	0.06	4.13	0.000
M2–DP (b2)	−0.06	−0.19	0.07	0.06	4.13	0.000
Total indirect effect	−0.09	−0.15	−0.04	0.02		
IV–m1–DV (a1b1)	−0.06	−0.11	−0.01	0.03		
IV–m2–DV (a2b2)	−0.06	−0.09	0.03	0.03		

Note—These are the path coefficients for the parallel mediation model of Hayes process model 4, indirect effects and 95% confidence interval predicting fear of COVID-19 (*n* = 301), SE is standard error, IV: independent variable (perceived control over time), DV: dependent variable (fear of COVID-19), m1 and m2: parallel mediators (mental health and vitality); a1, a2, b1, and b2 are regression coefficients for X1 and X2, respectively; while b1 and b2 are the regression coefficients for m1 and m2, respectively. Boot LL-CI and Boot UL-CI are abbreviations for the lower limit bootstrap confidence interval and upper limit bootstrap confidence interval.

## Data Availability

The data presented in this study are available on request from the corresponding author.

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
