# Peer review of "Do Mental Health and Vitality Mediate the Relationship between Perceived Control over Time and Fear of COVID-19? A Survey in an Italian Sample"

_jcm, 2021, doi:10.3390/jcm10163516_

Round 1
Reviewer 1 Report
It' s ok.
Author Response
Dear reviewer,
We are grateful for your appreciation of the scientific quality of our manuscript.
Sincerely
Barbara Caci and co-authors
Reviewer 2 Report
Dear Authors,
the paper "Do Mental Health and Vitality mediate the Relationship between Perceived Control over Time and Fear of COVID-19? A Survey in an Italian Sample" raises a crucial problem in nowadays situation, which is mental health at the time of Covid-19. The paper provides a useful insight to understand the impact of the pandemic on the psychological quality of life.
The introduction of the paper is meaningful, the methodology is consistent with the aims and hypotheses, the data analysis is well conducted, the results are connected with relevant publications.
I do recommend the paper suitable for publication.
Congratulation for this great publication!
Author Response

(The authors gave the same response as above.)

Reviewer 3 Report
Dear Authors,
the manuscript is well written and brings new knowledge to COVID-19 related mental health.
Minor points,
The prevalence and incidence of COVID-19 in the area of ​​study during the period of study should be discussed.
It is worth including in the introduction the following items relating to patients with chronic diseases, including people with autoimmune diseases, i.e. people whose mental health in the COVID-19 era requires attention.
Wańkowicz, P .; Szylińska, A .; Rotter, I. Evaluation of Mental Health Factors among People with Systemic Lupus Erythematosus during the SARS-CoV-2 Pandemic. J. Clin. Med. 2020, 9, 2872. https://doi.org/10.3390/jcm9092872
Wańkowicz, P .; Szylińska, A .; Rotter, I. The Impact of the COVID-19 Pandemic on Psychological Health and Insomnia among People with Chronic Diseases. J. Clin. Med. 2021, 10, 1206. https://doi.org/10.3390/jcm10061206
Wańkowicz, P .; Szylińska, A .; Rotter, I. Insomnia, Anxiety, and Depression Symptoms during the COVID-19 Pandemic May Depend on the Pre-Existent Health Status Rather than the Profession. Brain Sci. 2021, 11, 1001. https://doi.org/10.3390/brainsci11081001
Best regards
Author Response
We are very grateful to the reviewer for their suggestions. Below we provide point to point our answers to the reviewer’s comments.
#1 The prevalence and incidence of COVID-19 in the area of ​​study during the period of study should be discussed.
We discussed the prevalence and incidence of COVID-19 in lines 35-37 of our revised manuscript.
#2 It is worth including in the introduction the following items relating to patients with chronic diseases, including people with autoimmune diseases, i.e. people whose mental health in the COVID-19 era requires attention.
Wańkowicz, P .; Szylińska, A .; Rotter, I. Evaluation of Mental Health Factors among People with Systemic Lupus Erythematosus during the SARS-CoV-2 Pandemic. J. Clin. Med. 2020, 9, 2872. https://doi.org/10.3390/jcm9092872
Wańkowicz, P .; Szylińska, A .; Rotter, I. The Impact of the COVID-19 Pandemic on Psychological Health and Insomnia among People with Chronic Diseases. J. Clin. Med. 2021, 10, 1206. https://doi.org/10.3390/jcm10061206
Wańkowicz, P .; Szylińska, A .; Rotter, I. Insomnia, Anxiety, and Depression Symptoms during the COVID-19 Pandemic May Depend on the Pre-Existent Health Status Rather than the Profession. Brain Sci. 2021, 11, 1001. https://doi.org/10.3390/brainsci11081001
We cited the requested references in line 42.
Reviewer 4 Report
Dear Authors,
The submitted paper is focused on clarification of association between Perceived Control over Time and Fear of COVID-19 as mediated by mental health or vitality. Most interestingly, the authors are applying the specific statistical analysis of the mediation modeling as often reported in the psychological researches. I believe that a result of the mediation analysis could suggest various factorial effects including direct and indirect effects in the mediation modeling. However, since this statistical methodology is not commonly recognized in Clinical Medicine researches, almost of medical experts, except for psychologists who can apply the mediation analysis, may be difficult to interpret a result of the mediation modeling yielded in the current study. Considering the above issue, I would like to suggest that the authors need to explain in detail about the procedure of the mediation modeling and discuss carefully or clearly about interpretation to the mediation modeling. Consequently, I kindly suggest a major revision to the submitted manuscript.
- To my knowledge, the parallel mediation modeling has been reported by Baron and Kenny (1986) and is mainly applied in psychological researches. I believe that other experts except for psychology may be not able to understand methodological procedure of the mediation analysis. So, authors need to clearly mention about procedure of the parallel mediation modeling as the follows;
0) In theory, please explain about what you can hypothesize mediations of X→M→Y
1) Most importantly, please ensure a significant relationship between X and Y (X→Y).
2) After making the Path model of X→M→Y and X→Y, please estimate the Path Coefficients for each model.
3) Please make sure a significance of indirect effects of X→M→Y.
4) Please confirm how the effect of X→Y is changed compared with the mediation model of “1.Most importantly, please ensure a significant relationship between X and Y (X→Y).
Reference)
Baron RM, Kenny DA. The moderator-mediator variable distinction in social psychological research: conceptual, strategic, and statistical considerations. J Pers Soc Psychol. 1986;51(6):1173-1182.
- In previous researches, how a scale of Perceived Control over Time is applied? The below information cited is focused on student’s time management. If possible, the information available on application of the Perceived Control Over Time towards spread of infectious diseases should be mentioned in Discussion.
Reference)
The Perceived Control Over Time subscale by the Time Management Behaviour Scale of Macan et al. (28. Macan, T. H., Shahani, C., Dipboye, R. L., & Phillips, A. P. College student's time management: correlation with academic performance and stress. J Educ Psychol., 1990, 82, 760–768. http://doi.org/10.1037/0022-0663.82.4.760)
- Like a scale of Perceived Control over Time, previous findings regarding a relatively new scale of the Fear of COVID-19 should be included, with similar references applied the same scale.
- In the section of introduction, the authors raise five hypotheses consisting of H1 – Perceived control over time is negatively associated with COVID-19 fear, H2- Perceived control over time is negatively associated with adverse mental health statuses, H3- Perceived control over time is positively associated with vitality, H4- Adverse mental health statuses are positively associated with COVID-19 fear and H5- Vitality is negatively associated with COVID-19 fear. In order to promote understanding for medical experts who see the mediation analysis for the first time, the authors need to explain clearly about which kind of path (a1, a2, b1, b2, c or c’ as indicated in the figure) each hypothesis (from H1 to H5) supports with in Figure 2 within the section of Discussion.
- In so far references cited in the discussion, the statistical values (coefficients estimated in modeling or statistics yielded in the correlation analysis) have not been included. To compare statistics in the current study with those in previous studies, information on study design, the sample size and statistical data conducted in previous studies should be summarized or referred in the Discussion. Additionally, statistical values (coefficient, 95%CI, and p value) yielded by the mediation analysis of the present study need to be documented simply in the abstract.
Author Response
We are very grateful to the reviewer for suggestions. Below we provide point to point our answers to the reviewer’s comments.
#1 The submitted paper is focused on clarification of association between Perceived Control over Time and Fear of COVID-19 as mediated by mental health or vitality. Most interestingly, the authors are applying the specific statistical analysis of the mediation modeling as often reported in psychological researches. I believe that a result of the mediation analysis could suggest various factorial effects, including direct and indirect effects in the mediation modeling. However, since this statistical methodology is not commonly recognized in Clinical Medicine researches, almost of medical experts, except for psychologists who can apply the mediation analysis, it may be difficult to interpret a result of the mediation modeling yielded in the current study. Considering the above issue, I would like to suggest that the authors need to explain in detail the procedure of the mediation modeling and discuss carefully or clearly about interpretation to the mediation modeling. Consequently, I kindly suggest a major revision to the submitted manuscript.
#1 We are thankful to the reviewer for leading us to revise the manuscript carefully and improve its scientific quality and readability for the Journal audience.
#2 To my knowledge, parallel mediation modeling has been reported by Baron and Kenny (1986) and is mainly applied in psychological researches. I believe that other experts except for psychology may not be able to understand the methodological procedure of the mediation analysis. So, authors need to clearly mention about the procedure of the parallel mediation modeling as the follows;
0) In theory, please explain about what you can hypothesize mediations of X→M→Y
#2 We better specify the theoretical assumption of our model in line 167.
1) Most importantly, please ensure a significant relationship between X and Y (X→Y).
#2 We better specify the significant relationship between X and Y in lines 211-212
2) After making the Path model of X→M→Y and X→Y, please estimate the Path Coefficients for each model.
#3 We reported the estimated path coefficients for each model in paragraph 3.3.
3) Please make sure a significance of indirect effects of X→M→Y.
#3 We reported the indirect effects in paragraph 3.4.
4) Please confirm how the effect of X→Y is changed compared with the mediation model of “1.Most importantly, please ensure a significant relationship between X and Y (X→Y).
#2 We compared the direct and indirect effects in paragraph 3.5. We also follow the reviewer's suggestion and specify along with the paper the relative phases and paths.
Reference
Baron RM, Kenny DA. The moderator-mediator variable distinction in social psychological research: conceptual, strategic, and statistical considerations. J Pers Soc Psychol. 1986;51(6):1173-1182.
#3 In previous researches, how a scale of Perceived Control over Time is applied? The below information cited is focused on student’s time management. If possible, the information available on application of the Perceived Control Over Time towards spread of infectious diseases should be mentioned in Discussion.
Reference)
The Perceived Control Over Time subscale by the Time Management Behaviour Scale of Macan et al. (28. Macan, T. H., Shahani, C., Dipboye, R. L., & Phillips, A. P. College student's time management: correlation with academic performance and stress. J Educ Psychol., 1990, 82, 760–768. http://doi.org/10.1037/0022-0663.82.4.760)
#3 We inserted a reference for the Perceived Control Over Time application towards the spread of the COVID-19 pandemic in lines 297-299.
Like a scale of Perceived Control over Time, previous findings regarding a relatively new scale of the Fear of COVID-19 should be included, with similar references applied the same scale.
# 3 We inserted a reference for the prior applications of FEAR of COVID-19 scale in lines 333-334.
#4 In the section of introduction, the authors raise five hypotheses consisting of H1 – Perceived control over time is negatively associated with COVID-19 fear, H2- Perceived control over time is negatively associated with adverse mental health statuses, H3- Perceived control over time is positively associated with vitality, H4- Adverse mental health statuses are positively associated with COVID-19 fear and H5- Vitality is negatively associated with COVID-19 fear. In order to promote understanding for medical experts who see the mediation analysis for the first time, the authors need to explain clearly about which kind of path (a1, a2, b1, b2, c or c’ as indicated in the figure) each hypothesis (from H1 to H5) supports with in Figure 2 within the section of Discussion.
# 4 We reported each path associated with the appropriate hypothesis in the Discussion section.
#5 In so far references cited in the discussion, the statistical values (coefficients estimated in modeling or statistics yielded in the correlation analysis) have not been included. To compare statistics in the current study with those in previous studies, information on study design, the sample size and statistical data conducted in previous studies should be summarized or referred in the Discussion.
#5 Even if we are conscious that reporting coefficients estimated in modeling or statistics yielded in the correlation analysis in the Discussion would be more informative, it could also limit the readability of our manuscript. Indeed, Tables and Figures report all the data, as well as the Result section. Considering that our study is a cross-sectional one and the model allows us to make not causal inferences, we retain more appropriate not compare our statistics with that of prior studies.
#6 Additionally, statistical values (coefficient, 95%CI, and p value) yielded by the mediation analysis of the present study need to be documented simply in the abstract.
# 6 We provided the requested information in the Abstract.
Round 2
Reviewer 4 Report
Dear Authors,
The resubmitted manuscript has been revised carefully according to the reviewer's comments.
This manuscript is a resubmission of an earlier submission. The following is a list of the peer review reports and author responses from that submission.
Round 1
Reviewer 1 Report
I thank the authors for engaging in this important research, which I commend: I enjoyed reading it and I hope my comments below will be useful for improving further their manuscript.
Line 18: age as subscript (Mage)
Keywords: could be chosen by including terms not mentioned in the title, so as to make it easier to search for the paper
Introduction: it must be implemented and focused more considering the psychological variables discussed in the article (eg. Vitality)
Line 59-61: Where does this assertion find reference in the literature?
Materials and methods: This section is described in a sufficiently clear way, if possible, please specify better the sample recruitment in procedure section.
Line 168: APA citation not Vancouver
Results:
Line 200: Reporting of p-value should never be reported = 0.000
The use of capital p value (P) is unfamiliar. The authors should standardize the abbreviation
Preliminary results section is too confusing, should be reorganized (eg. sub-paragraphs)
Line 281: APA citation not Vancouver
The Discussions section are well done, but the Conclusions are not immediately focused on the main themes of the paper.
Best regards.
Reviewer 2 Report
Authors show an useful MS in the actual context of COVID. Nevertheless, in the last year, there are many MS published similar to this. In the case of this MS, statistic analysis are original comparing to other. However, sample procedure used is not rigorous. So, I have doubts about the replicability of results obtained by the authors.